# Gesture Generation with Diffusion Models Aided by Speech Activity Information

Rodolfo L. Tonoli*†
r105652@dac.unicamp.br
Department of Computer Engineering
and Automation, School of Electrical
and Computer Engineering,
University of Campinas (UNICAMP)
Campinas, SP, Brazil

Leonardo B. de M. M. Marques*
Lucas H. Ueda
lmenezes@cpqd.com.br
lhueda@cpqd.com.br
CQPD
Campinas, SP, Brazil

Paula D. P. Costa†
paulad@unicamp.com
Department of Computer Engineering
and Automation, School of Electrical
and Computer Engineering,
University of Campinas (UNICAMP)
Campinas, SP, Brazil

## ABSTRACT

This paper describes a gesture generation model based on state-of-the-art diffusion models. Novel adaptations were introduced to improve motion appropriateness relative to speech and human-likeness. Specifically, the main focus was to enhance gesture responsiveness to speech audio. We explored using a pre-trained Voice Activity Detector (VAD) to obtain more meaningful audio representations. The proposed model was submitted to the GE-NEA Challenge 2023. Perceptual experiments compared our model, labeled SH, with other submissions to the challenge. The results indicated that our model achieved competitive levels of human-likeness. While appropriateness to the agent's speech score was lower than most entries, there were no statistically significant differences from most models at the confidence level.

## CCS CONCEPTS

• **Computing methodologies** → **Animation**; *Intelligent agents*; Machine learning.

## KEYWORDS

Gesture generation, co-speech gestures, diffusion models

**ACM Reference Format:**
Rodolfo L. Tonoli, Leonardo B. de M. M. Marques, Lucas H. Ueda, and Paula D. P. Costa. 2023. Gesture Generation with Diffusion Models Aided by Speech Activity Information. In *INTERNATIONAL CONFERENCE ON MULTIMODAL INTERACTION (ICMI '23 Companion), October 9–13, 2023, Paris, France*. ACM, New York, NY, USA, 7 pages. https://doi.org/10.1145/3610661.3616554

## 1 INTRODUCTION

Human communication is composed of verbal and nonverbal behaviours. Co-speech gestures are one of these behaviours. They are

---

*Both authors contributed equally to this research.

†Also with Artificial Intelligence Lab., Recod.ai, Institute of Computing, University of Campinas, SP, Brazil..

---

visible actions of any body part produced while speaking and may serve different purposes, such as to provide emphasis or to depict some physical property [30]. Being such a key part of human communication, gestures are employed in embodied agents to simulate real interactions and create believable characters [29]. Otherwise, these agents may be perceived as lifeless or dull.

Recent research focused on automatic gesture generation (or synthesis) through deep learning. Such systems are able to animate embodied agents much faster and less time-demanding than traditional techniques such as hand-crafted animations or motion capture. Additionally, these techniques may not be suited for applications whose speech content is unknown beforehand, such as an avatar being controlled by a human or an embodied agent powered by a language model.

Most research on gesture generation has a cross-modal mapping approach to this problem, similar to a translation between different behaviour modalities [4]. Also, gestures are correlated with prosody and may be associated with semantics [21]. Thus, most systems use speech audio, speech text, or both to guide gesture generation [23].

However, synthetic data still struggles to appear human-like and appropriate to speech if compared to real human data [33]. More challenging scenarios could widen the gap between synthetic and real data. For example, in dyadic interactions, people are expected to take turns being the active speaker for brief or long moments. Most research has not addressed such situations. We propose a monadic gesture generation model that considers the voice activity for better alignment and responsiveness of gestures given speech audio. The model is based on a composition of the DiffuseStyleGesture [32], a speech-driven diffusion model and the Motion Diffusion Model (MDM) [26], which is text-driven. The main contributions of this paper to the aforementioned models are:

- the integration of voice activity information to improve turn-taking and speech audio synchrony while using only monadic inputs;
- the employment of aligned speech text as input through a pre-trained CLIP model, thus supporting the generation of gestures semantically related to speech;
- the use of speech audio representations suited for content-related tasks from a pre-trained WavLM model.

Our code can be accessed via https://github.com/AI-Unicamp/ggvad-genea2023.

This article is structured as follows: Section 2 presents related works on gesture generation and diffusion; the data processing is

detailed in Section 3; Section 4 describes the proposed model and qualitative evaluations of our model are presented in Section 5; the results of the proposed model compared to other entries to the GENEA Challenge 2023 are detailed in Section 6; and Section 7 presents the conclusion and final remarks.

## 2 BACKGROUND AND PRIOR WORK

Generative models enable the capture of the one-to-many nature of gestures. Studies using VAEs [9], GANs [8], and Normalizing Flows [10] show that such models surpass deterministic ones. However, these approaches still suffer from generalized problems such as mean pose convergence and training instability. Recently, diffusion models arose as a new class of promising generative models achieving state-of-the-art results across a wide range of multimodal tasks validated by perceptual evaluations without the same pitfalls as the generative models mentioned before. Additionally, these models were shown to be capable of handling data with special structures, efficient sampling and providing improved likelihood estimation [31].

Denoising Diffusion Probabilistic Models (DDPMs) [12] are a type of generative model that synthesize new samples from an underlying data distribution by learning how to reconstruct information. During the training process, the model takes one noisy data point ($x_t$), obtained by applying $t$ Gaussian noise addition steps to the original data ($x$), with $0 < t \leq T$, as $T$ is the size of the complete diffusion noise-adding chain, and is set to equivalently predict either a one-step denoised sample ($x_{t-1}$), a fully reconstructed data point ($x_0$), or the noise contained ($\epsilon$). On inference, the process is started from a pure Gaussian noise distribution and the reconstruction is performed iteratively $T$ times, generating a new sample [12].

Diffusion models exhibited state-of-the-art performance in several different tasks. On image synthesis, diffusion models achieved superior performance to the at the time GAN-based state-of-the-art synthesis [7], and were also proven to be able to generate and edit hyper-realistic images [22, 25]. In the audio domain, diffusion models have been successfully exploited for audio generation [15] and text-to-audio [19] tasks, obtaining higher performance when compared to other current staple models. Recently, diffusion models have also been explored on the task of video generation, which were demonstrated to synthesize high-fidelity videos with a high degree of controllability and world knowledge [11].

In the context of human motion generation, text-based models aim to control the movements via natural language semantically. The MotionDiffuse model [35] is the first model to exploit DDPMs for this task, combining these models with a cross-modal Transformer based architecture. In another approach, denominated Motion Diffusion Model (MDM) [26], textual representations extracted from a pre-trained CLIP [24] are combined with a Transformer model in a classifier-free guidance diffusion training process [13]. Other works tackle the dance generation task, which intends to generate dances given music as audio input. The EDGE [27] method pairs a diffusion model with Jukebox, a generative model for music, whereas the Listen, Denoise and Action! [1] model adapts DiffWave [15] to generate poses and synthesize dances in various styles.

More recently, diffusion models have also been applied to the gesture generation task. DiffMotion [34] is the first approach that applies DDPMs to generate gestures. It leverages an autoregressive temporal encoder based on an LSTM that processes context represented by spectral audio features and previous poses to condition a diffusion process, generating each pose individually.

The DiffGesture [37] model uses a convolutional audio encoder to extract representations directly from the raw audio. A Transformer model then uses these representations that undergoes an implicit classifier-free guidance diffusion training.

The GestureDiffuCLIP [2] model introduces a multimodal (text, motion or video) prompt-conditioned style-controlled gesture generation via mode-specific pre-trained CLIP encoders. Also, they use a contrastive learning strategy to learn semantic correspondences between textual transcripts of the input speech and gestures, allowing for the generation of semantically-aware gestures. These contributions, along with a denoiser network based on Transformers, attention, and AdaIN layers [14] to incorporate style guidance, compose a latent diffusion training process [25].

Finally, the DiffuseStyleGesture [32] model combines layers of cross-local and global attention to better capture the localized aspects of gestures. With representations extracted from the self-supervised WavLM model [6], the authors perform a diffusion training process and are able to generate and control gestures based on a style label.

Although the increasing interest in the field, the synthesized motions from most models are still far from being indistinguishable from real human motion [33]. Moreover, research often concentrates on monadic scenarios in which only one participant actively communicates. Consequently, crucial behaviours of real-life interactions, such as listening, reciprocal expression, and interruptions, are disregarded during development and evaluation.

## 3 DATA AND DATA PROCESSING

The dataset used by the 2023 GENEA Challenge is an adaptation of the Talking With Hands 16.2M (TWH) data [18]. Pre-processing, data augmentation, and selection are described in the challenge's main paper [17]. The available dataset presents a dyadic scenario, i.e., it is composed of data from two people having a conversation, referred to as the main agent and interlocutor. Entries to the challenge should only generate movements for the main agent, and using the interlocutor's data was optional. Available data includes motion, speech audio, speech text (audio transcripts with timestamps), and speaker label. We only used data from the main agent; thus, our model depends on monadic information alone despite the dyadic scenario. Speaker labels were also ignored.

The dataset motions are BVH files with movements composed of 30 poses per second represented by Euler angles. We extracted each pose and composed a feature vector $g = [\rho_p, \dot{\rho}_p, \rho_r, \dot{\rho}_r]$ where $\rho_p \in \mathbb{R}^{3j}$ and $\dot{\rho}_p \in \mathbb{R}^{3j}$ are the global 3D joint positions and positional velocities, $\rho_r \in \mathbb{R}^{6j}$ and $\dot{\rho}_r \in \mathbb{R}^{3j}$ are the local 6D joint rotations [36] and the local 3D joint rotational velocities, $j$ represents the number of joints. The 30 frames per second rate of the original data and all 83 joints of the skeleton were preserved, thus $g \in \mathbb{R}^{1245}$ for each pose. Each dimension of motion data is normalized to zero mean and unit standard deviation over the

challenge training set. Audio files were resampled from 44.1 kHz to 16 kHz.

## 4 METHOD

Our approach consists of a combination of the MDM [26] and the DiffuseStyleGestures [32] models, with modifications aiming for improved responsiveness of gestures given speech audio. The architecture is shown in Figure 1. Our model generates sequences of 120 poses simultaneously, corresponding to 4 seconds. We consider inputs to be divided into global and fine-grained information. The first corresponds to information relevant to the 4-second sequence as a whole, which includes the words spoken (text), seed poses, and timestep embedding. On the other hand, fine-grained information is considered to be relevant at the frame level; thus, it includes audio and speech activity.

### 4.1 Global Information

Since gestures can be semantically related to speech, providing text information could improve gesture appropriateness. As textual features, we use spoken words within a motion sequence. Words timestamps from the audio transcript are used for extracting the corresponding words. As in the MDM [26] model, the speech text contained in the sequence of poses passes by a pre-trained CLIP [24] model[1] and then processed from the clip output dimension of 512 to a dimension of 64 by a fully connected layer.

For the motion between consecutive generated sequences to have cohesion, 10 previous seed poses are used as conditional input. These poses are flattened and then projected to a dimension of 192, and then concatenated with the textual information, forming a vector with the defined latent dimension of 256. Additionally, the timestep embedding of the diffusion process, which indicates which denoising step is being performed, is a sinusoidal positional embedding that is passed through two fully connected layers with a Sigmoid Linear Unit (SiLU) activation layer in between and projected to latent dimension. With this, the embedding that represents global conditioning information (the one that is invariant to the pose sequence) is obtained by summing the time-step embedding with the concatenation of the textual and seed poses embedding.

### 4.2 Fine-grained Information

We work with chunks of sequences of 120 poses corresponding to 4 seconds of motion. The noisy poses for the diffusion process are obtained by adding $t$ steps of Gaussian noise on a sequence. These poses are then projected via a linear layer from the pose dimension of 1245 to the latent space dimension. For the audio information, we use the resampled audio data and pass it through the WavLM [6] model[2]. Differently from the DiffuseStyleGestures [32], we use the representations extracted from the 11th layer instead of the 12th. The 11th layer is reported to perform better at content-related tasks, such as phoneme recognition and automatic speech recognition. These representations are first interpolated to match the length of the corresponding pose sequence and then projected to a dimension of 64 by a linear layer.

---

[1]Version 'ViT-B/32' obtained from https://github.com/openai/CLIP
[2]Version 'Base+' obtained from https://github.com/microsoft/unilm/tree/master/wavlm

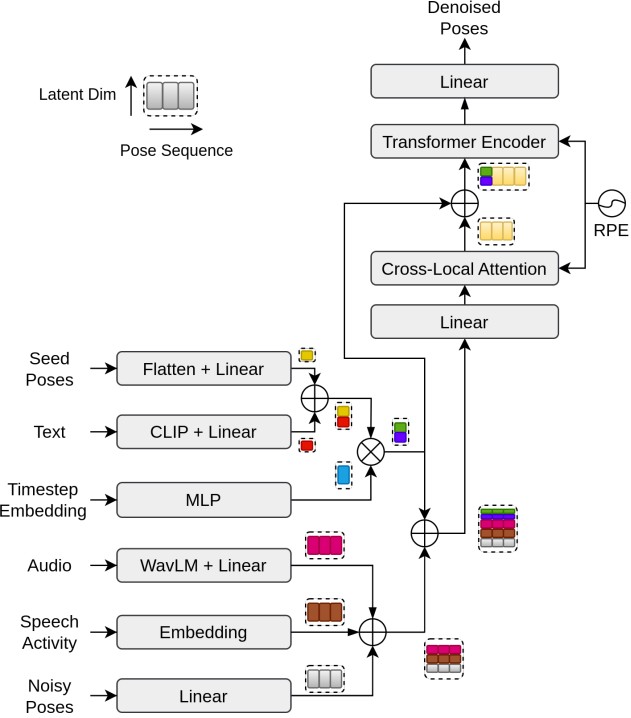

**Figure 1: Model architecture.**

*4.2.1 Speech Activity Information.* Due to the dyadic nature of the dataset, some sections of the data are composed of moments in which the main agent is not the active speaker, such as listening and turn-taking moments. Gestures performed in active or non-active moments may play different roles in human interaction and, thus, differ from those performed in other moments. For example, *beat* gestures occur during articulations of speech and may serve to emphasize what is being said [21]; differently, mimicry, often performed automatically, may enhance helpfulness and strengthen social bonds [28]. Although our model only uses monadic data, we introduce the use of speech activity information. This information, otherwise embedded in audio representations such as spectrograms and MFCCs, may be lost in the abstract WavLM representations. Furthermore, the interpolation of representations to match the pose sequence can blend moments with and without speech activity. Thus, the contribution of such inclusion is believed to be two-fold. First, it provides more straightforward access to fine-grained speech energy. Second, it helps to stress, during training, the difference between gestures in the aforementioned moments, not in terms of functionality, but dynamics.

Speech activity can be inferred through analytical approaches such as energy and F0. However, the dataset audios contain noise that could affect computing these parameters: various speakers, different speech volumes, and background noise such as speech from the interlocutor and breathing. Thus, we consider two scenarios for acquiring speech activity information. The first is based on a pre-trained Voice Activity Detector (VAD)[3] that consists in

---

[3]Obtained from https://huggingface.co/speechbrain/vad-crdnn-libriparty

a small CRDNN (A combination of convolutional, recurrent and deep neural network) trained on the Libriparty dataset[4], which is a synthetic cocktail-party scenario derived from the Librispeech dataset. When speech is detected, the model outputs a 1 and otherwise a 0. The second approach is taken from the annotated speech text timestamps provided in the dataset. When there is any text, we consider the respective timestamps as 1 and otherwise as 0.

The major difference between these approaches is that the pre-trained model can detect intra-text pauses, whereas audio transcripts provide word-level timestamps granularity. A comparison of both is shown in Figure 2. From the figure, it is noticeable that VAD provides closer alignment with speech energy. Besides, the pre-trained VAD removes the need for audio-aligned annotated speech text, which is sensitive to human perception or error.

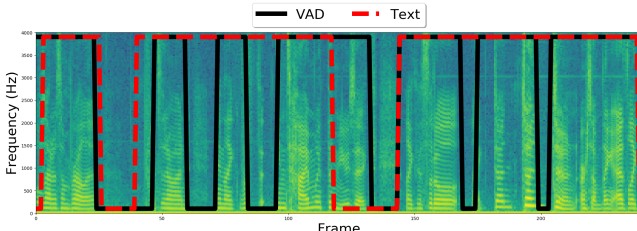

**Figure 2: Scaled speech activities from timed audio transcripts (red) and from the VAD (black) overlapped with a spectrogram of an eight-second audio sample in the background.**

The speech information sequence extracted from the VAD is used to select two embeddings with latent dimensions representing the presence of speech or no speech for each pose. This sequence of embeddings is then concatenated with the noisy poses and the audio embeddings forming the fine-grained information.

### 4.3 Training

The fine-grained information is concatenated with the global information along the latent dimension. Then, all the input information is projected back to the latent dimension by an input linear layer and fed to the cross-local attention layer to capture local relations between the features. Then, we concatenate the global information embedding one more time with the output along the sequence dimension before passing the sequence to the transformer encoder to capture the global context. Then, we ignore the first token of the output sequence and project the outputs to the pose dimension, which finally represents the denoised pose ($x_0$) itself. We use positional embeddings to add sequence information on both the cross-local attention and the transformer encoder.

On inference, a sequence at a time is generated. The model outputs a vector $G = [g_1, g_2, \cdots, g_{120}]$. The last 10 poses from the previously generated sequence are used to condition the generation of the next sequence; mean poses are used for conditioning the first sequence.

---

[4]https://github.com/speechbrain/speechbrain/tree/develop/recipes/LibriParty/generate_dataset

For post-processing, we use linear interpolation to impose continuity between successive sequences. To smooth motion artifacts in the output, we also apply a Savitzky-Golay [20] filter with a window length of 9 and polynomial order of 3.

The model was trained for 290k steps, with a batch size of 64, in a single NVIDIA Titan Xp GPU, which took about 2.5 days.

## 5 EVALUATION

There still is no objective metric to measure gesture perception reliably. Moreover, previous research has found that object metrics differ from subjective ones [16]. Therefore, the research team empirically evaluated the proposed model, its variations, and the reference models through visual inspection of their outputs.

We trained the MDM [26] and the DiffuseStyleGestures [32] and used them as references for comparison, i.e., a starting point for development. Although providing reasonable human-like motion, in terms of appropriateness to speech, we found the results unsatisfactory. The outputs seemed unaware of moments such as brief pauses, turn-taking, and listening moments. That is, the agent would frequently make gestures in those moments that appeared inadequate and similar to behaviours performed when it was the active speaker. So, our main focus in developing the model for the GENEA Challenge 2023 was to overcome those issues of disregard for no-speak moments. Thus, a VAD was employed to leverage speech activity information.

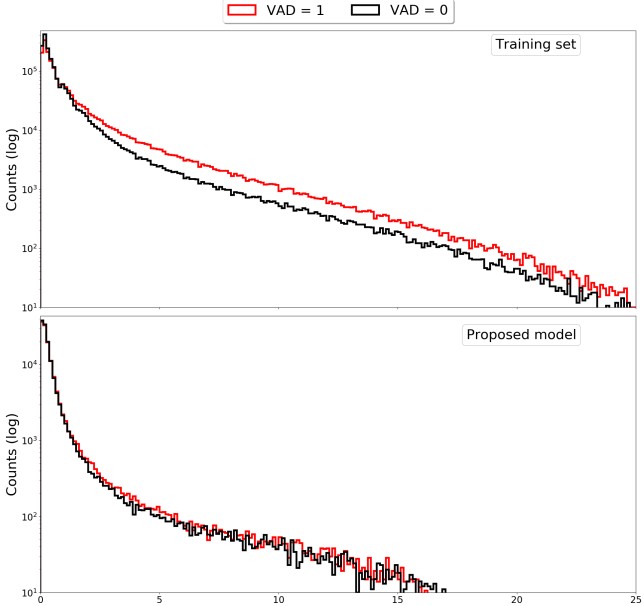

**Figure 3: Histograms of the rotational velocities from the main agent's left and right forearm joints from the training set of the dataset (top), and the output of the proposed model with (bottom). Red and black indicate velocities extracted when the main agent was the active speaker and when it was not.**

In order to examine the effectiveness of the VAD, we present histograms of the rotational velocities of the forearms, a joint that is very active when gesturing with the arms, on Figure 3, for the real training set (top), and the output of the proposed model (bottom). The figure splits each set considered in two distributions: when the VAD indicates that there is an occurrence of speech, VAD output equals one, and when the VAD indicates that there is no speech, its output is zero.

For the training set, the histograms reveal distinct patterns associated with speech activity during gesticulation. Speakers in the dataset exhibit increased forearm movements while talking versus silence periods. These insights support our underlying assumption that people tend to perform more gestures - or at least more abrupt gestures - when they are speaking.

The proposed model could reproduce, to some extent, the overall behaviour of the training set. However, it was unable to synthesize motion that reproduced the differences seen in the training set given speech activity, that is, a larger concentration of higher velocities when the agent is speaking. We did an ablation study with the proposed model without the VAD module. Its histogram was similar to the one with VAD. However, visual inspections of the outputs by the research team favored outputs by the proposed model with VAD in terms of speech and gesture alignment.

We also compared outputs from models with and without text input. However, we did not find a significant amount of semantically related gestures in their output. Further investigation should be carried out to indicate if there is a sufficient amount of such gestures in the dataset for models to be able to learn from. Still, we kept texts as input as motion quality was not impaired.

Compared to our reference models, the output of the proposed model seems better, especially in terms of speech audio and gesture alignment. However, we notice that some artifacts are still present in the motions. Motions occasionally converge to an unusual or odd-looking pose, absurd rotations still take place, and jittering is sometimes noticeable.

## 6 RESULTS AND DISCUSSION

The results of the shared evaluations of the GENEA Challenge 2023 indicated that our model (condition SH) is competitive with most conditions in terms of human-likeness but obtained relatively poor results for appropriateness to speech [17].

Figure 4 presents human-likeness ratings. Subjects participants gave their ratings based on how human-like the motions appeared, from 0 (worst) to 100 (best). Real motion data (NA) achieved a median rating of 71, the baselines 46 (BD) and 43 (BM), while our condition scored 46. We believe that the module that contributed the most to the human-likeness of generated gestures is the attention mechanism. As Yang et al. [32] showed in their ablation studies, the cross-local attention module played a significant role in terms of human-likeness ratings.

Two evaluations were performed to assess gesture appropriateness to speech: appropriateness for agent speech and for the interlocutor speech. The first contains mainly moments where the main agent is the active speaker, while the roles are reversed in the latter.

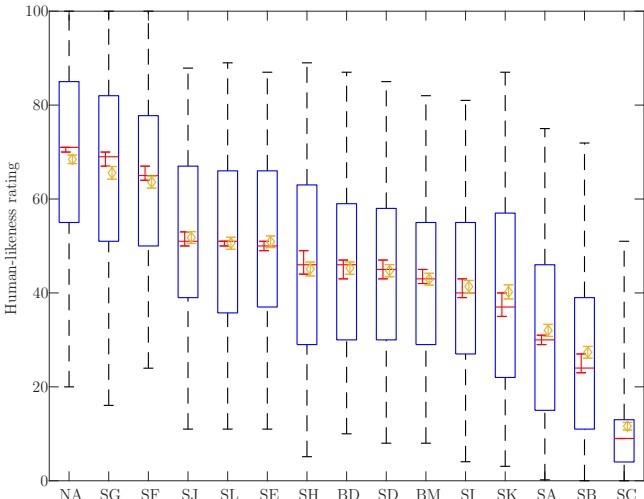

**Figure 4: Shared human-likeness rating study. Red bars are median ratings; yellow diamonds are mean ratings. Entries to the challenge are labeled SA-SL (ours is SH), BD and BM are the baselines [5], and NA is real data. Extracted from Kucherenko et al. [17].**

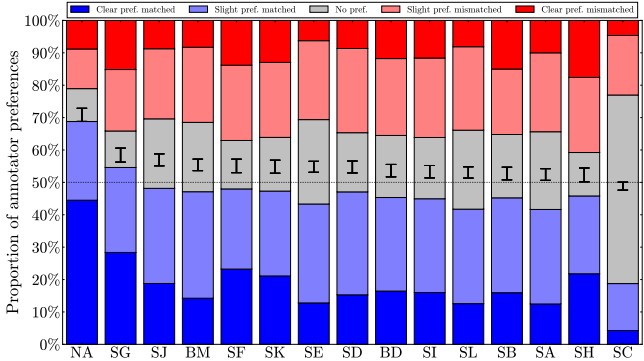

**Figure 5: Shared appropriateness for agent speech responses. Entries to the challenge are labeled SA-SL (ours is SH), BD and BM are the baselines [5], and NA is real data. Extracted from Kucherenko et al. [17].**

For the appropriateness of agent speech evaluation, subjects were presented with speech audio and two motions generated by the model. One motion is the output generated with the speech audio presented as input, and the other is the output from another segment of speech audio. For our condition, subjects preferred the matching motion 52.9% of the time, slightly above chance. Although one of the lowest mean appropriateness scores, there is no statically significant differences in the scores of ours and another ten conditions (conditions BM to SA, in Figure 5).

Our condition had the lowest score in the appropriateness for the interlocutor evaluation. This means that subjects found the mismatched stimuli more appropriate. However, our model does not use any interlocutor information as input. Thus, from the model's

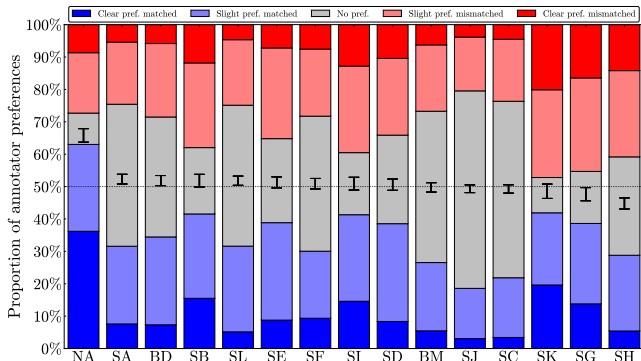

**Figure 6: Shared appropriateness for the interlocutor responses. Entries to the challenge are labeled SA-SL (ours is SH), BD and BM are the baselines, and NA is real data. Extracted from Kucherenko et al. [17].**

perspective, the output of both matched and mismatched stimuli for this evaluation was generated using the same inputs. Evidently, it is not expected that the outputs be exactly the same due to the probabilistic nature of the model. But both outputs are expected to be equivalent in terms of human-likeness and appropriateness, thus scoring similarly to chance (50%).

We also noticed from all three evaluations that our model had a wide range of scores. For instance, whiskers from the box plot visualization of Figure 4 span almost the entire $y$-axis; our condition, along with condition SK, had the highest confidence intervals of median and mean ratings. In the appropriateness for agent speech, our condition had the third highest number of clear preferences for matched stimuli, the highest for mismatched, and the second lowest for no preferences when compared to other entries to the challenge. Thus, we argue that the proposed model is indeed capable of generating gestures that are competitive in terms of human-likeness and appropriateness for the main agent. However, the artifacts mentioned in the previous section hinder gesture perception and should be addressed before any conclusion regarding the proposed architecture and individual modules.

## 7 CONCLUSION

This paper describes the proposed diffusion-based model for gesture generation that uses pre-trained VAD. Incorporating speech activity information in such models could improve responsiveness during rapid back-and-forth interactions. Also, a VAD can explicitly provide this information without needing human-annotated transcripts, thus potentially suited for real-time dialogue. Our model has been compared with others in the GENEA Challenge 2023, a crowdsourced evaluation that directly compares different methods while controlling factors such as data and evaluation methodology. The evaluation showed that our model is compatible with other entries to the challenge in terms of human-likeness, but appropriateness to speech is still unimpressive despite our efforts.

Our experiments revealed mixed results regarding the effectiveness of the proposed implementation improvements to the gesture generation system. While convergences to undesired poses, extreme

joint rotations, and jittering were not frequent, they nonetheless occurred. Besides, output motion was unstable, i.e., when generating motions given the same inputs, the resulting motion quality varied greatly. These issues may have contributed to subpar performance in evaluations and compromised the responsiveness of generated gestures to speaking moments. Although our adaptations hold potential value for gesture generation tasks, further improvements are needed to leverage their benefits fully. Especially the explicit use of speech activity information that could be leveraged to address turn-taking moments

We intend to focus primarily on improving speech and gesture alignment for future work. An interesting approach is adapting an external framework for alignment as the one proposed by Badlani et al. [3]. Another obvious path is to incorporate data from the interlocutor to capture the aspects of dyadic scenarios.

## ACKNOWLEDGMENTS

This study was partially funded by the Coordenação de Aperfeiçoamento de Pessoal de Nivel Superior – Brasil (CAPES) – Finance Code 001. The first author is grateful to the Eldorado Research Institute.

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
