# OpenReview forum: "Gesture Generation with Diffusion Models Aided by Speech Activity Information"
_ACM.org/ICMI/2023/Workshop/GENEA_Challenge — GENEA Challenge 2023 Workshopproceeding_

### Official Review · Reviewer_cCbX · 2023-07-22
**This paper clearly describes the details and the clear reason of using VAD to improve speech-gesture synchronization.**

**Rating:** 6
**Confidence:** 4

**Review:**

The paper is clear and concise. It uses VAD as the additional input to improve synchronization. As there remains a gap between natural and synthetic gestures, the study of features is important for improving speech appropriateness. The weakness of this paper, however, is that dyadic settings are not described.

The followings are some suggestions:
1. It would be great if the authors can have a figure like Fig. 3, but showing the velocity and voice activity along time for a gesture. It would strongly justify the selection of VAD features. It's interesting to see to what extent voice activity correlates with the velocity strengths of an arm joint.

2. As far as I know, the baselines use energy and timed text transcripts for fine-grained information. Energy or prosody features seem to provide richer information than binary voice activity detections. Adding comparisons of these features would be very helpful.

3. Consider adding discussions of the voice activity of the interlocutor. How could it affect the main agent's gesture?

4. Please cite the baseline, "The ivi lab entry to the genea challenge 2022".

---

### Official Review · Reviewer_vhoP · 2023-07-29
**The main advantage of this work is the innovative use of diffusion models and relevant auxiliary information to improve the correlation and semantic coherence between speech and gestures, leading to more natural and appropriate gesture generation.Although the results are fair, the analysis is detailed. I recommend accepting it after modification.**

**Rating:** 6
**Confidence:** 5

**Review:**

Abstract:

    The article presents a gesture generation method based on diffusion models. The proposed approach utilizes a pre-trained Voice Activity Detector (VAD), meaningful audio representations, and textual information to generate responsive and natural-looking gestures. The method was evaluated in the GENEA Challenge 2023 and achieved fair results concerning human-likeness. Overall, the proposed method showcases innovation in the field of gesture generation and has the potential to enhance the naturalness and appropriateness of generated gestures.

Review Feedback:

（1）Please explain in detail the meaning of Figure 2 in the text and explain why only VAD was finally used.

（2）Although the network uses speech activity indicator model to identify speech interruptions, the gestures may not necessarily stationary correspond to those periods of silence, potentially confusing the network. Is it possible to consider adjusting the gestures at VAD=0 in the data preprocessing.

（3）From Figure 3, it can be observed that the occurrences of vad=0 at offsets like 80 and 100 did not decrease significantly compared to the occurrences of vad=1. I would like to know if this satisfies the author's original intention in observations, such as the absence of movement during pauses.

（4）Compared to the original disunion model, does the inclusion of this additional information result in objective and subjective improvements? This is not explicitly demonstrated in the article.

（5）The author utilizes a training approach where sequences of 120 poses are trained at once. How was this segmentation considered? Does this segmentation (grouping every 120 frames) disrupt the temporal coherence of the original data?

I hope the above feedback can assist you in improving your work.

---

### Decision · Program_Chairs · 2023-08-04

**Decision:**

Accept (Workshop proceeding)

**Comment:**

Congratulations! All of the reviewers recommended accepting this paper. Reviewers appreciated the innovative use of diffusion models. The organisers have decided to accept your paper to the Workshop ICMI track to be published in the Adjunct ICMI Proceedings.
We suggest that the authors carefully consider the feedback received from the reviewers and use it to improve their manuscript for the challenge camera-ready submission deadline. Below follows some input from the organisers based on the paper and the reviews:
1. An important statement in the abstract is false: “the appropriateness for agent speech results is not statistically different than most entries and both baselines”. The statistical analysis showed that the proposed model SH was statistically worse than the baseline BM in appropriateness to agent’s speech.
2. Discuss in more detail the usage of VAD instead of energy or prosody features.
3. Discuss how the modifications from the original diffusion model have influenced the results.
4. Provide more details about Figures 2 and 3.
5. Cite the baseline method, as the code was used for data processing.